# *NRG1* Gene Fusions—What Promise Remains Behind These Rare Genetic Alterations? A Comprehensive Review of Biology, Diagnostic Approaches, and Clinical Implications

**DOI:** 10.3390/cancers16152766

**Published:** 2024-08-05

**Authors:** Tomasz Kucharczyk, Marcin Nicoś, Marek Kucharczyk, Ewa Kalinka

**Affiliations:** 1Department of Pneumonology, Oncology and Allergology, Medical University of Lublin, 20-059 Lublin, Poland; marcin.nicos@umlub.pl; 2Department of Zoology and Nature Conservation, Institute of Biology, Maria Curie-Sklodowska University in Lublin, 20-033 Lublin, Poland; marek.kucharczyk@mail.umcs.pl; 3Oncology Clinic, Institute of the Polish Mother’s Health Center in Lodz, 93-338 Lodz, Poland; ewakalinka@wp.pl

**Keywords:** NSCLC, *NRG1* fusions, molecularly targeted

## Abstract

**Simple Summary:**

Neuregulin-1 (NRG1) is an important regulator of ErbB-mediated pathways involved in cancer development. Recently, there have been several studies analyzing *NRG-1* gene fusions engaged in altering the dimerization of HER family proteins and the consecutive results of their activation in different types of cancer. Non-small cell lung cancer (NSCLC) patients can benefit from pan-HER inhibitors, and knowledge of *NRG-1* fusions can help tailor the treatment to a specific group of patients. New drugs targeting cells with *NRG-1* fusions are under clinical trials and show effectiveness in NSCLC treatment.

**Abstract:**

Non-small cell lung cancer (NSCLC) presents a variety of druggable genetic alterations that revolutionized the treatment approaches. However, identifying new alterations may broaden the group of patients benefitting from such novel treatment options. Recently, the interest focused on the neuregulin-1 gene (*NRG1*), whose fusions may have become a potential predictive factor. To date, the occurrence of *NRG1* fusions has been considered a negative prognostic marker in NSCLC treatment; however, many premises remain behind the targetability of signaling pathways affected by the *NRG1* gene. The role of *NRG1* fusions in ErbB-mediated cell proliferation especially seems to be considered as a main target of treatment. Hence, NSCLC patients harboring *NRG1* fusions may benefit from targeted therapies such as pan-HER family inhibitors, which have shown efficacy in previous studies in various cancers, and anti-HER monoclonal antibodies. Considering the increased interest in the *NRG1* gene as a potential clinical target, in the following review, we highlight its biology, as well as the potential clinical implications that were evaluated in clinics or remained under consideration in clinical trials.

## 1. Introduction

In the era of precision therapy, novel driver alterations are extensively studied to qualify the patient for the best-fitting treatment. In 2020, lung cancer accounted for 11.9% of all new cancer diagnoses in Europe, constituting about 480,000 people [1]. Non-small cell lung cancer (NSCLC), which includes 85% of lung cancer cases, is one of the cancer types that presents a variety of actionable genetic changes, with quite a few available targeted drugs that have revolutionized the treatment approaches [2]. Nevertheless, the percentage of patients that receive already popular epidermal growth factor receptor (EGFR) tyrosine kinase inhibitors (TKIs), anaplastic lymphoma kinase (ALK) inhibitors, or less common molecules is still low. The search for new targets is still ongoing to allow a larger group of patients treatment tailored specifically for them [3]. Identifying such alterations or fusions may effectively select those benefiting from such novel treatment options. Apart from gene mutations, rearrangements and fusions of different genes are among the most commonly diagnosed cancer cell driver alterations [2]. The inhibitors of anaplastic lymphoma kinase (*ALK*)*,* ROS proto-oncogene 1 (*ROS1*), rearranged during transfection (*RET*), and neurotrophic tyrosine receptor kinase (*NTRK)* rearrangements are already present in our everyday clinical practice [4,5]. Recently, interest has focused on neuregulin-1 (NRG1) as a potential oncogenic target.

The *NRG1* gene harbors several variants that have been classified on the Evidence for Sequence-variant Classification (ESCAT) scale as likely benign (rs3924999—intron region; rs7832768—promoter region) or uncertain significance (rs10503929 and rs16879552—both intronic), and their clinical relevance should be confirmed [6]. On the other hand, *NRG1* fusions might be considered the main oncogenic factor in solid tumors. The first description of such fusions (*CD74-NRG1*) in invasive mucinous adenocarcinoma of the lung (IMA) was in 2014. The targeted drugs for patients with *NRG1* fusions are still in clinical trials, and the search continues [7].

*NRG1* rearrangements are uncommon compared to other more often described gene alterations found in NSCLC. In one of the studies, it was present in 0.5% of patients (2 of 404 analyzed cases) [8], in the other in 0.3% of patients (25 of 9252 analyzed samples) [9]. The prevalence of *NRG1* rearrangements in other types of cancers is similar to NSCLC and amounts to 0.5% in cholangiocarcinoma, pancreatic carcinoma, and renal cell carcinoma, 0.4% in ovarian cancer, and 0.2% in breast cancer and sarcoma [7,9,10]. Thus, the NRG1 fusions may be considered as biomarkers in various cancer types. Moreover, these fusions exclude the occurrence of other cancer-driving genetic changes. In some NSCLC cases, however, their presence was described along with mutations in *KRAS* and *BRAF* or *ALK* rearrangements [9,11].

To date, the occurrence of *NRG1* fusions was considered a negative prognostic marker in NSCLC treatment. Patients harboring such alterations presented reduced overall survival (OS) when treated with standard chemotherapy, chemoimmunotherapy, or immunotherapy alone. However, there are many premises behind the targetability of pathways affected by this alteration [9,12,13].

Considering the increased interest in the *NRG1* gene as a potential clinical target, this review discusses the structure and biology of the *NRG1* gene and the occurrence of potential genetic fusions. Furthermore, we indicate the *NRG1* fusion detection methods that are based on both high-throughput or single-gene approaches. In the end, we highlight the druggable applications of *NRG1* fusions as the first or secondary target in the treatment of NSCLC, which are already available in clinics or are still under consideration in clinical trials.

## 2. Structure and Biology of NRG1 Fusions

Neuregulin 1 is a protein, encoded by the *NRG1* gene located on the short arm of chromosome 8 (8p12), that is involved in various biological processes, including neural development, synaptic plasticity, myelination, and inter-cell signaling in the heart and breast [14]. The function of NRG1 is necessary for the early stages of development, and its absence, as was shown in mouse models, does not allow for proper embryonic development [15]. *NRG1* gene has many tissue-specific isoforms, created through alternative splicing, that differ structurally from each other. However, most isoforms contain the same extracellular epidermal growth factor-like (EGF-like) domain [14,16,17], which is crucial in the case of *NRG1* fusions to keep the functionality of the aberrant protein and drive cancer cell development. Most isoforms of NRG1 are bound to the cell membrane as a precursor. During proteolytic processes, the mature NRG1 is released, which can be transported further from the cell of origin and activate receptors on the surface of other cells. However, isoform III of neuregulin 1 retains the EGF-like domain in the membrane, which allows for the activation of mainly neighboring cells [18]. Moreover, there are some premises that epigenetic changes may also dysregulate the NRG1 expression, leading to its involvement in cancer development and progression [19]. The schematic localisation and structure of the *NRG1* gene is presented in Figure 1.

The EGF-like domain of NRG1 protein is mainly an activator of Erb-B2 tyrosine kinase receptor 3 (ErbB3 also called HER3, human epidermal growth factor receptor 3), subsequently activating heterodimerization, most frequently ErbB2-ErbB3, but also EGFR or ErbB4, and further downstream signaling through mitogen-activated protein kinase (MAPK) and phosphoinositide-3-kinase (PI3K)/AKT/mammalian target of rapamycin (mTOR) pathways [20,21]. Although ErbB3 has seriously decreased kinase activity, its dimerization with other ErbB family receptors, after activation by NRG1, allows further downstream activation of the aforementioned pathways [22,23]. The NRG1 fusion proteins act as abnormal activators of ErbB-mediated cell proliferation pathways, and the result of such activation is the promotion of proliferation of molecularly altered cells.

It is postulated that *NRG1* fusions act similarly to neuregulin-1 isoform III, with a membrane-attached EGF-like domain. Different *NRG1* fusions can activate different homo- or heterodimers of ErbB [24], hence, they can activate diverse downstream pathways and result in alternative results from blockade attempts. The scheme of dimeric ErbB downstream signaling pathways regulated by the NRG1 protein is presented in Figure 2.

## 3. Occurrence of *NRG1* Fusions

As previously stated, the first discovery of the *CD74-NRG1* fusion was described in 2014 in a study of 25 lung adenocarcinoma patients without *KRAS* or *EGFR* mutations. The described five cases were detected in non-smoking females with the IMA subtype [7]. Since that discovery, most of the *CD74-NRG1* fusion cases have been presented in this subtype of lung adenocarcinoma [25]. Subsequently, other groups of researchers identified different fusion partners of the *NRG1* gene: *SLC3A2* [9,12,26], *SDC4* [9,27], *RBPMS* [9,28], *VAMP2* [29], *WRN* [28], *ATP1B1* [27], *ROCK1* [25], *RALGAPA1* [30], *TNC* [9], *MDK* [9], *DIP2B* [9], *MRPL13* [9], *DPYSL2* [9], *PARP8* [9], *THAP7* [25], *SMAD4* [25], *KIF13B* [13], *ITGB1* [31], *UBXN8* [10], *NPTN* [32], *CADM1* [33], *F11R* [33], *FGFR1* [33], *FLYWCH1* [33], *KRAS* [33], *PLCG2* [33], and *VAPB* [33]. To date, the study by Jonna and co-workers is the most comprehensive analysis regarding *NRG1* fusions in solid tumors, where the incidence of the most common partners was as follows: *CD74* (29%), *ATP1B1* (10%), *SDC4* (7%), and *RBPMS* (5%). All the other fusions detected in the analyzed group of 21,858 solid tumor samples occurred with 2% frequency [9].

Fusions of *NRG1* and a few different partner genes have been described in other tumors as well, namely in ovarian cancer: *SETD4* [9], *TSHZ2* [9], *ZMYM2* [9], *RAB3IL1* [25], and *CLU* [25,34], and in pancreatic ductal adenocarcinoma: *VTCN1* [9], *CDH1* [9], *CDH6* [35], *SARAF* [10,35], *APP* [36], and *CDK1* [37]. Other described individual cases include breast cancer: *ADAM9* [9], *COX10-AS1* [9], *AKAP13* [25], *FOXA1* [25], *DDHD2* [10], *FUT10* [10], *BRE* [10], *CD9* [10], *ARHGEF39* [38], *FAM91A1* [38], and *ZNF704* [38], colorectal carcinoma: *IKBKB* [10], *ZCCHC7* [10], *TNRFSF10B* [10], *ERO1L* [10], and *KCTD9* [10], esophageal carcinoma: *BIN3* [10] and *CCAR2* [10], gallbladder carcinoma: *NOTCH2* [9], head and neck squamous carcinoma: *THBS1* [25] and *PDE7A* [25], bladder cancer: *GDF15* [9], renal cell carcinoma: *PCM1* [25], prostate carcinoma: *STMN2* [25] and *UNC5D* [39], neuroendocrine tumor of the nasopharynx: *HMBOX1* [9,40], spindle cell sarcoma: *WHSC1L1* [9] and *PPHLN1* [40], as well as uterine carcinosarcoma: *PMEPA1* [25]. Interestingly, in fusions of *NRG1* and *PCM1*, *STMN2*, and *PMEPA1*, the EGF-like domain was not observed; hence, the functionality and the activating ability of these fusions may not be relevant as oncogenic drivers [25]. The overview of locations of *NRG1* fusion gene partners in different tumors within chromosome 8 and other chromosomes is presented in Table 1 and Figure 3 and Figure 4, respectively.

## 4. Detection of *NRG1* Fusions

The rare occurrence of *NRG1* fusions requires a robust detection method, especially in a wide range of partner genes. The main obstacle is to capture all probable fusions in a single sample using an economically viable tool.

The comprehensive identification of *NRG1* fusions may involve next-generation sequencing (NGS) technologies, both RNA and DNA-based, which allow for high-throughput genomic profiling of tumor samples [8,41]. Since the gene spans over 1Mb, NGS allows for full analysis of all *NRG1* alterations, those known and unknown as well [9]. RNA-based sequencing can be used to identify fusions located in-frame, allowing for detecting products of transcription from alternative splicing forms, which are common in the case of the *NRG1* gene [8,9]. The most useful method in such an approach would be whole transcriptome sequencing (WTS), as it allows the detection of all possible transcripts. However, the drawback of WTS is that the method needs high-quality RNA isolated from the sample [28,42]. The DNA-based NGS approaches, whole exome sequencing (WES) and targeted sequencing allow, on the other hand, for the description of exact sequences of breakpoints but do not tell if these sequences undergo translation [9]. Such methods also do not cover intronic sequences properly, which is a disadvantage, as the *NRG1* gene consists of large, non-coding fragments that might carry the possible breaking points [16]. Another drawback of the DNA-based NGS approach is the poor quality of DNA extracted from formalin-fixed paraffin-embedded (FFPE) tissue samples that, in the computational analysis, may deliver a high number of artifacts that may imitate the false positive results [43,44]. However, due to the mentioned limitations and high costs of NGS-based approaches, other single gene-based methods are still of great interest for *NRG1* gene detection.

The most common technique used for the detection of fusion genes and their protein products is immunohistochemistry (IHC). The method is relatively fast, cheap, and provides high sensitivity and specificity, although it needs a qualified pathologist for proper description [45]. It was postulated that phosphorylated ErbB3 (pErbB3) protein analysis could be the first step to identifying tumors carrying *NRG1* fusions [21,33]. The association between high pErbB3 expression detected with IHC in IMA and non-IMA lung cancer samples was shown by Trombetta et al. [46]. The main issue with IHC is that it might show false-positive results, as it presents fusion proteins that undergo full expression (transcription and translation) [47], hence, the method can be used mainly as the first step in screening and selection of samples for further, more complex analysis [9].

The third approach to the detection of *NRG1* fusions is the fluorescence in situ hybridization (FISH) technique [48]. It is commonly available in most molecular laboratories, but it requires more expertise and experience from the diagnostician when interpreting the results. It is more labor-consuming and works well with previously described fusions. Also, the technique cannot describe specific breakage points in fusion partners [36,46]. Besides IHC and FISH techniques, real-time PCR and Sanger sequencing also allow the detection of the exact known genomic breakpoints but remain underused and are very limiting [9]. On the other hand, Nanostring technology may become the RNA-based approach that will allow efficient estimation of the level of expression of all the exons in the region of interest within the *NRG1* gene [49].

## 5. *NRG1* Fusion as the Predictive Factor in Lung Cancer Treatment

The activation of signal transduction by binding of NRG1 ligand to ErbB family receptors or the process of ErbB family protein dimerization is considered the main target of treatment in patients harboring *NRG1* fusions [50,51]. Hence, NSCLC patients harboring *NRG1* fusions may benefit from targeted therapies such as HER family inhibitors, which have shown efficacy in previous studies in various cancers. The first choice in such an approach would be afatinib. This irreversible pan-HER inhibitor was proven effective in NSCLC patients harboring *EGFR* gene-activating mutations [52,53]. Several studies analyzed the effectiveness of afatinib in patients harboring *NRG1* fusions, although they were mainly case studies. Drilon et al. reported no response to afatinib treatment in four patients with IMA histology, although there were visible results in patient-derived xenograft mouse models [25]. Gay et al. presented two cases of lung cancer patients without *EGFR* mutations, carriers of *SLC3A2-NRG1* and *CD74-NRG1* fusions. The patients received afatinib with documented durable responses of 10 and 12 months, respectively [8]. Another case study of five lung cancer patients harboring *CD74-NRG1* or *SDC4-NRG1* fusions, treated with afatinib, resulted in four cases of partial response (PR) (5–27 months) and one stable disease (SD) (4 months) [54]. On the other hand, a single-patient case study with a *CD74-NRG1* fusion presented by Wu et al. indicated that afatinib showed PR for seven months until the progression of the disease [55]. A larger study by Liu et al. with different types of tumors included 29 NSCLC patients treated with afatinib, and it showed a 48.3% overall response rate (ORR), including three complete responses (CRs) and eleven PRs, with a median duration of response (DoR) of 6.8 months and median PFS of 6.1 months [56].

Tarloxotinib, another small molecule pan-ErbB inhibitor, in a hypoxic tumor environment decreased the phosphorylated ErbB-related process by targeting the membrane of reductase STEP4 protein, leading to tumor growth inhibition and cancer regression. The results were observed in patient-derived cell lines and multiple murine xenograft models harboring an *NRG1* fusion [9,57,58]. Apart from the small-molecule pan-ErbB inhibitors mentioned above, there are also many positive premises behind the inhibition of NRG1-related pathways by monoclonal antibodies binding to the ErbB receptors. Odintsov et al. reported that seribantumab (anti-ErbB3 antibody, MM-121/SAR256212) decreased activation of the PI3K-AKT, mTOR, and ERK pathways in *NRG1* fusion-positive patient-derived lung and breast cancer cell lines and patient-derived xenograft (PDX) models from lung and ovarian cancer patients. Moreover, seribantumab efficiently blocked other ErbB family members, indicating a similar to afatinib reduction of proliferation and induction of apoptosis [59]. In the end, Drilon et al. observed durable tumor regression in a PDX mouse model and anti-proliferative activity in the MDA-MB-175-VII cell line [25]. The summary of the effect of different drugs on the prognosis of lung cancer patients is presented in Table 2.

## 6. *NRG1* Fusion as a Secondary Target in Lung Cancer Treatment

*NRG1* gene fusions have the potential to affect the activity of ErbB-related pathways; thus, from one side, there is a potential treatment option for cancer patients harboring *NRG1* fusion-positive cancers by HER-targeted therapies [60]. However, some studies have indicated that *NRG1* fusion drives the primary resistance to molecularly targeted therapies by activation of the HER3 [61] and HER3/AKT [62] signaling pathway. Due to the complexity of the ErbB-related signal transduction pathways, the NRG1-driven resistance has the potential to be overcome by the application of treatment regimens based on multi-targeted agents. For instance, trastuzumab combined with anti-HER3 monoclonal antibody, pertuzumab, or poziotinib may revert the resistance process in cell lines [61,62,63].

In NSCLC, *NRG1* fusions are listed as acquired oncogenic alterations associated with the acquired resistance to EGFR-TKIs driven by activation of the NRG1/ErbB3 pathway [64,65]. Moreover, it was also shown that *ALK*-rearranged NSCLC cells acquire resistance to ALK inhibitors, losing the *EML4/ALK* fusion and activating the NRG1/ErbB3 pathway [9]. In such a situation, the sensitivity to crizotinib may be restored by pan-ErbB inhibitors, afatinib or dacomitinib, in the absence of other secondary *ALK* mutations [66]. In case of resistance to alectinib, the NRG1/ErbB3 activation maintains survival and stimulates mesenchymal activity, driving the epithelial–mesenchymal transition (EMT) that is the main hallmark of cancer dissemination [67,68]. This phenomenon may be confirmed by the observation that high expression of ErbB3 and NRG1 significantly correlated with brain metastases from primary lung tumors [69].

## 7. Clinical Trials Related to Patients with Solid Tumors Harboring *NRG1* Fusions

Besides the published results, there are also some clinical trials evaluating the targeted treatment possibilities in patients harboring *NRG1* fusions (Table 3). The clinical trial NCT03805841 [70] evaluated the ORR to tarloxotinib in NSCLC patients harboring insertion in exon 20 of the *EGFR* gene, activating mutation of *HER2* or *NRG1* fusion. However, the study was terminated, and the outcome has not been provided yet. The efficient blocking of ErbB family members by seribantumab was confirmed in metastatic cancer patients having high and low levels of NRG1 and ErbB2 expression, respectively (NCT01447706 [71], NCT01151046 [72], NCT00994123 [73]). Moreover, in the CRESTONE study (NCT04383210) in the cohort of NSCLC patients harboring *NRG1* fusions who received seribantumab, the ORR and the disease control rate were 39% and 94%, respectively. The overall duration of response ranged from 1.4 to 17.2 months [74,75].

Further, Zenocutuzumab (MCLA-128), a bispecific monoclonal antibody against ErbB2 and ErbB3, in the eNRGy study (NCT02912949) demonstrated durable efficacy and a well-tolerated safety profile in patients with advanced solid tumors harboring *NRG1* fusion, regardless of tumor histology [76]. Moreover, the NCT01966445 trial showed that GSK2849330, an anti-ErbB3 monoclonal antibody, elicited a durable 19-month response in NSCLC patients harboring the *CD74-NRG1* fusion. In the end, poziotinib in the ZENITH20 clinical trial (NCT03318939) demonstrated antitumor activity with a durable response and manageable safety profile as the second-generation TKI in previously treated NSCLC patients with *HER2* exon 20 insertions [77,78,79].

## 8. Conclusions and Future Perspectives

The application of deep sequencing techniques, such as next-generation sequencing, has provided a wide array of data about the molecular background of NSCLC and opened routes for treatment personalization. This advancement revolutionized the management of therapy in these deadly conditions. Moreover, the studies shed light on how rare alterations affect the signaling pathways, indicating they impact treatment response or acquire resistance to targeted approaches. To date, the occurrence of *NRG1* fusions, which is a very rare alteration in solid tumors, was considered a negative prognostic marker in NSCLC treatment; however, gaining knowledge about its impact on ErbB signaling pathways has provided significant attention in recent scientific research, offering a potential avenue for targeted therapy. Recent and ongoing clinical trials and preclinical studies have explored the effectiveness of both already available and new agents in *NRG1* fusion-positive lung cancers, demonstrating promising results in terms of response rates and disease control. Thus, including such NSCLC cases in planning treatment regimens is reasonable. Moreover, re-evaluating standard approaches to NSCLC molecular analysis to detect possibly actionable, novel gene fusions or alterations that may affect well-known signaling pathways seems relevant and shows promise for further clinical improvement.

## Figures and Tables

**Figure 1 cancers-16-02766-f001:**
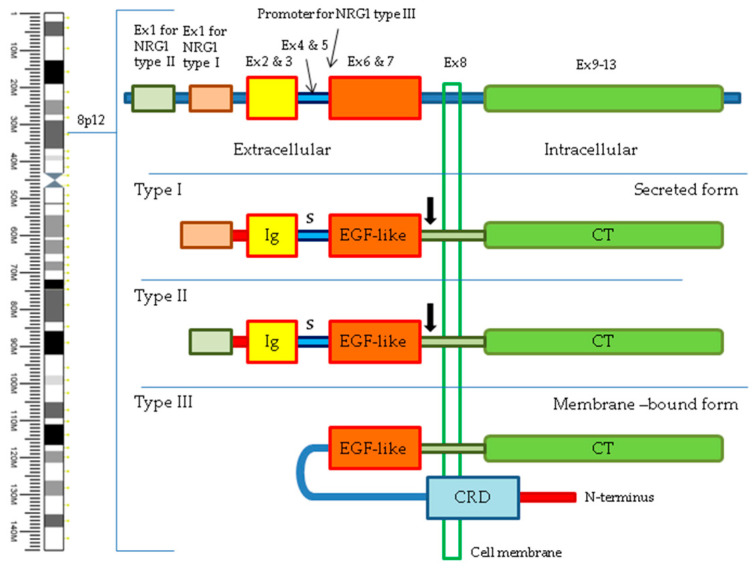
Schematic of *NRG1* gene structure showing position on chromosome 8, composition of exons, and structure of protein isoforms. Ig—immunoglobulin-like domain; S—stalk; EGF-like—EGF-like domain; CT—cytoplasmic tail; CRD—cysteine-rich domain. Black arrows indicate the location of cleavage in secreted types of NRG1.

**Figure 2 cancers-16-02766-f002:**
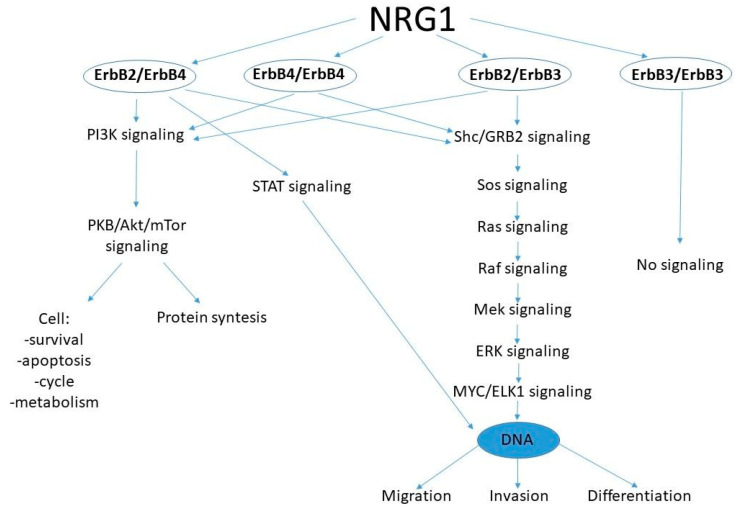
The scheme of dimeric ErbB downstream signaling pathways regulated by the NRG1 protein. The NRG1 protein, through different biological cascades, affects the ErbB dimers for protein synthesis, cell survival, cell apoptosis, control of cell cycle and metabolism, as well as cell migration, invasion, or differentiation.

**Figure 3 cancers-16-02766-f003:**
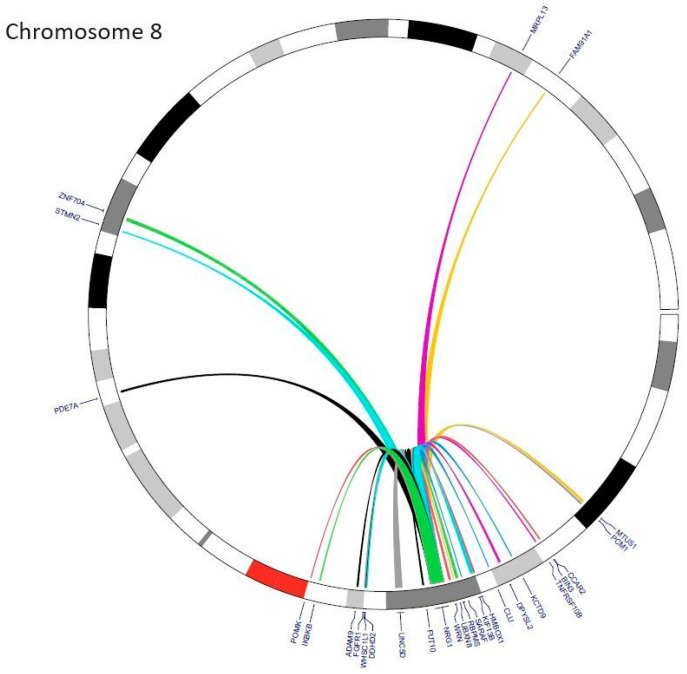
The circos plot presents the common fusion partners of *NRG1* within chromosome 8.

**Figure 4 cancers-16-02766-f004:**
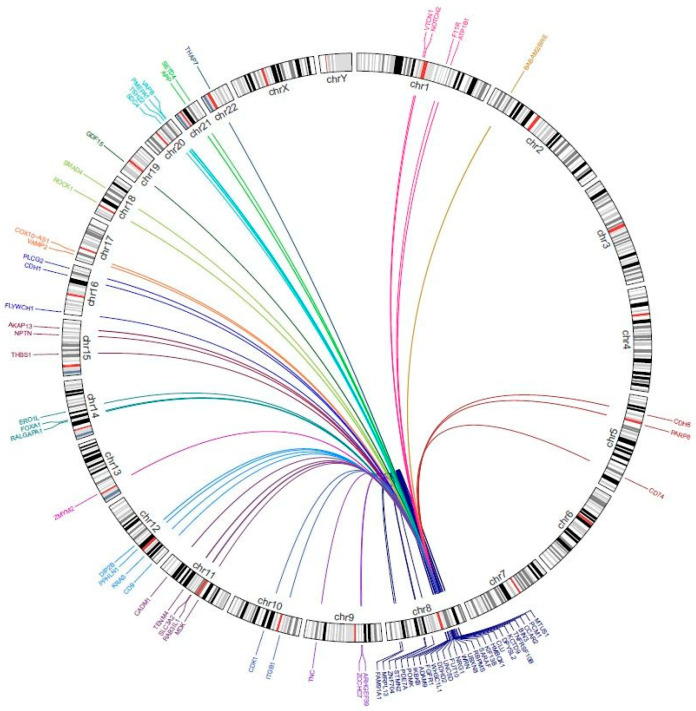
The circos plot presents the common fusion partners of NRG1 with genes localized within all chromosomes.

**Table 1 cancers-16-02766-t001:** Partner genes, with their chromosomal localization and translocation description, including the *NRG1* gene, in different cancer types.

Fusion Gene	Localization	Aberration	Cancer Type
ADAM9	8p11.22	t(8;8)(p12;p11)	Breast Cancer
AKAP13	15q25.3	t(8;15)(p12;q25)	
ARHGEF39	9p13.3	t(8;9)(p12;p13)	
BABAM2/BRE	2p23.2	t(8;2)(p12;p23)	
CD9	12p13.31	t(8;12)(p12;p13)	
COX10-AS1	17p12	t(8;17)(p12;p12)	
DDHD2	8p11.23	t(8;8)(p12;p11)	
FAM91A1	8q24.13	t(8;8)(p12;q24)	
FOXA1	14q21.1	t(8;14)(p12;q21)	
FUT10	8p12	t(8;8)(p12;p12)	
TENM4	11q14.1	t(8;11)(p12;q14)	
ZNF704	8q21.13	t(8;8)(p12;q21)	
ATP1B1	1q24.2	t(8;1)(p12;q24)	Breast Cancer/Cholangiocarcinoma/Pancreatic Ductal Adenocarcinoma
ERO1L	14q22.1	t(8;14)(p12;q22)	
IKBKB	8p11.21	t(8;8)(p12;p11)	Colorectal Cancer
KCTD9	8p21.2	t(8;8)(p12;q21)	
POMK	8p11.21	t(8;8)(p12;p11)	
TNFRSF10B	8p21.3	t(8;8)(p12;p21)	
ZCCHC7	9p13.2	t(8;9)(p12;p13)	
BIN3	8p21.3	t(8;8)(p12;p21)	Esophageal Carcinoma
CCAR2	8p21.3	t(8;8)(p12;p21)	
NOTCH2	1p12	t(8;1)(p12;p12)	Gallbladder Cancer
PDE7A	8q13.1	t(8;8)(p12;q13)	Head and Neck Squamous Cell Carcinoma
THBS1	15q14	t(8;15)(p12;q14)	
PCM1	8p22	t(8;8)(p12;p22)	Kidney Renal Clear Cell Carcinoma
CD74	5q33.1	t(8;5)(p12;q33)	Lung Adenocarcinoma/Pancreatic Adenocarcinoma
CADM1	11q23.3	t(8;11)(p12;q23)	Lung Cancer
DIP2B	12q13.12	t(8;12)(p12;q13)	
DPYSL2	8p21.2	t(8;8)(p12;p21)	
F11R	1q23.3	t(8;1)(p12;q23)	
FGFR1	8p11.23	t(8;8)(p12;q11)	
FLYWCH1	16p13.3	t(8;16)(p12;p13)	
ITGB1	10p11.22	t(8;10)(p12;p11)	
KIF13B	8p12	t(8;8)(p12;p12)	
KRAS	12p12.2	t(8;12)(p12;p12)	
MDK	11p11.2	t(8;11)(p12;p11)	
MRPL13	8q24.12	t(8;8)(p12;q24)	
NPTN	15q24.1	t(8;15)(p12;q24)	
PARP8	5q11.1	t(8;5)(p12;q11)	
PLCG2	16q23.3	t(8;16)(p12;q23)	
RALGAPA1	14q13.2	t(8;14)(p12;q13)	
SDC4	20q13.12	t(8;20)(p12;q13)	
SLC3A2	11q12.3	t(8;11)(p12;q12)	
SMAD4	18q21.2	t(8;18)(p12;q21)	
THAP7	22q11.21	t(8;22)(p12;q11)	
TNC	9q33.1	t(8;9)(p12;q33)	
WRN	8p12	t(8;8)(p12;p12)	Lung Cancer/Breast Cancer
RBPMS	8p12	t(8;8)(p12;p12)	Lung Cancer/Renal Cell Carcinoma
HMBOX1	8p21.1-p12	t(8;8)(p12;p21)	Neuroendocrine Tumor of the Nasopharynx/Spindle Cell Sarcoma
CLU	8p21.1	t(8;8)(p12;p21)	Ovarian Cancer
RAB3IL1	11q12.2-q12.3	t(8;11)(p12;q12)	
SETD4	21q22.12	t(8;21)(p12;q22)	
TSHZ2	20q13.2	t(8;20)(p12;q13)	
ZMYM2	13q12.11	t(8;13)(p12;q11)	
APP	21q21.3	t(8;21)(p12;q21)	Pancreatic Adenocarcinoma
CDH1	16q22.1	t(8;16)(p12;q22)	
CDH6	5p13.3	t(8;5)(p12;p13)	
CDK1	10q21.2	t(8;10)(p12;q21)	
ROCK1	18q11.1	t(8;18)(p12;q11)	
SARAF	8p12	t(8;8)(p12;p12)	
UNC5D	8p12	t(8;8)(p12;p12)	
VTCN1	1p13.1-p12	t(8;1)(p12;p13)	
STMN2	8q21.13	t(8;8)(p12;q21)	Prostate Cancer
WHSC1L1	8p11.23	t(8;8)(p12;p11)	Sarcoma
MTUS1	8p22	t(8;8)(p12;p22)	Spindle Cell Sarcoma
PPHLN1	12q12	t(8;12)(p12;q12)	
GDF15	19p13.11	t(8;19)(p12;p13)	Urothelial Bladder Cancer
PMEPA1	20q13.31	t(8;20)(p12;q13)	Uterine Carcinosarcoma

**Table 2 cancers-16-02766-t002:** A summary of the effect of different drugs on the prognosis of lung cancer patients.

Drug	Studied Material	Effect	Reference
Afatinib	NSCLC patients	10–12 months of durable response	Gay et al. [8]
		5–27 months of partial response 4 months of stable disease	Cadranel et al. [54]
		7 months of partial response	Wu et al. [55]
		48.3% of the overall response rate 6.8 months median duration of response 6.1 months median progression-free survival	Liu et al. [56]
Tarloxotinib	patient-derived cell lines, murine xenograft models	inhibition of tumor growth cancer regression	Bhandari et al. [57] Estrada-Bernal et al. [58]
Seribantumab	patient-derived cell lines, patient-derived xenograft models	reduction of proliferation induction of apoptosis	Odintsov et al. [59]
	patient-derived xenograft mouse model MDA-MB-175-VII cell line	durable tumor regression anti-proliferative activity	Drilon et al. [25]

**Table 3 cancers-16-02766-t003:** A summary of clinical trials dedicated to patients with solid tumors (including NSCLC) harboring *NRG1* fusions. Data were collected from the ClinalTrials.gov database (http://clinicaltrials.gov/ (accessed on 30 July 2024)).

Clinical Trial ID (Duration) Status	Tested Drug (Phase)	Genetic Eligibility	Conditions (Cohort)	Primary Measured Outcomes
NCT05919537 (09.2023–03.2031) Recruiting	HMBD-001 with/without chemotherapy (Phase I)	*NRG1* fusions Extracellular domain HER3 mutations	Advanced solid tumors (68)	1. Adverse events 2. Incidence and nature of dose-limiting toxicities (DLTs) 3. ORR
NCT02912949 (01.2015–12.2026) Recruiting	Zenocutuzumab (MCLA-128) (Phase 2)	*NRG1* fusions	Solid tumors (250)	1. ORR 2. Duration of response
NCT04383210 (09.2020–03-2025) Active, not-recruiting	Seribantumab (Phase 2)	*NRG1* fusions	Locally advanced or metastatic solid tumors (75)	ORR
NCT04750824 (10.2020–12.2021) Completed	Afatinib (Observational)	*NRG1* fusions	Solid tumors (110)	ORR
NCT05057013 (11.2021–09.2026) Recruiting	HMBD-001 (Phase 1)	*NRG1* fusions HER3 expression	Solid tumors (135)	1. Recommended dose 2. Adverse events 3. ORR
NCT03805841 (03.2019–04.2021) Terminated	Tarloxotinib (Phase 2)	*NRG1* fusions *ERBB* family fusions *EGFR* Exon 20 Insertion *HER2*-activating mutations	NSCLC or advanced solid tumors (41)	ORR

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
