# Peer review of "NRG1 Gene Fusions—What Promise Remains Behind These Rare Genetic Alterations? A Comprehensive Review of Biology, Diagnostic Approaches, and Clinical Implications"

_cancers, 2024, doi:10.3390/cancers16152766_

Round 1
Reviewer 1 Report
Comments and Suggestions for Authors
The aim of this review was to highlight the biology ofthe NRG1 gene and its fusions, as well as to summarize the potential of its clinical implications that have been evaluated in clinics or are under consideration in clinical trials. This is a very interesting topic in oncology and the authors have clearly highlighted its importance. However, there are still some questions that need to be clarified.
3. Occurrence of NRG1 fusions
Please add a table presenting the data and listing the partner genes in different cancer cases.
5. NRG1 fusions as a predictive factor in the treatment of lung cancer
Please add a separate paragraph describing only the clinical trials that look at patients with solid tumors (including NSCLC) that have NRG1 fusions
5. NRG1 fusion as a predictive factor in the treatment of lung cancer
Add a table showing the effect of different drugs on the prognosis of lung cancer patients
Author Response
Reviewer 1
We are very grateful for the Reviewer’s constructive comments, which have greatly helped us restructure our manuscript. We hope the Reviewer will find our revised manuscript more precise and useful for the readership of Cancers
The aim of this review was to highlight the biology of the NRG1 gene and its fusions, as well as to summarize the potential of its clinical implications that have been evaluated in clinics or are under consideration in clinical trials. This is a very interesting topic in oncology and the authors have clearly highlighted its importance. However, there are still some questions that need to be clarified.
- Occurrence of NRG1 fusions
Please add a table presenting the data and listing the partner genes in different cancer cases.
Following the remark we summarized the data listing the partner genes in different cancer cases in Table 1.
- NRG1 fusions as a predictive factor in the treatment of lung cancer
Please add a separate paragraph describing only the clinical trials that look at patients with solid tumors (including NSCLC) that have NRG1 fusions
Following the remarks we provided the elaboration and a table regarding the clinical trials in a separate paragraph number 7
- NRG1 fusion as a predictive factor in the treatment of lung cancer
Add a table showing the effect of different drugs on the prognosis of lung cancer patients
Following the remark we provided in Table 2 a summary of the effect of the different drugs described in paragraph 5.
Reviewer 2 Report
Comments and Suggestions for Authors
The authors presented a review of the NRG1 gene fusion and its clinical application in the diagnosis and treatment in non-small cell lung cancer (NSCLC). This review is overall informative and intriguing. I have several minor comments:
1. In section “Structure and biology of NRG1 fusions”, it would be helpful to provide a schematic picture of NRG1 showing its position and composition of exons/isoforms. Because the authors also mentioned that isoform III of neuregulin 1 retains the EGF-like domain in the membrane.
2. In section “Detection of NRG1 fusions”, there is a recent method detecting fusion genes using whole-exome sequencing data in cancer patients (PMID: 35251131), would this algorithm help for the detection of NRG1 fusions? Please comment on this point.
3. All figures in this submission are blurred and the text is difficult to read, the authors should provide higher-resolution images. Ideally, they should use vector graphics, which will maintain their quality and readability across different viewing scales and formats.
Comments on the Quality of English Language
Minor editing of English language required
Author Response
Reviewer 2
We are very grateful for the Reviewer’s constructive comments, which have greatly helped us restructure our manuscript. We hope the Reviewer will find our revised manuscript more precise and useful for the readership of Cancers
The authors presented a review of the NRG1 gene fusion and its clinical application in the diagnosis and treatment in non-small cell lung cancer (NSCLC). This review is overall informative and intriguing. I have several minor comments:
- In section “Structure and biology of NRG1 fusions”, it would be helpful to provide a schematic picture of NRG1 showing its position and composition of exons/isoforms. Because the authors also mentioned that isoform III of neuregulin 1 retains the EGF-like domain in the membrane.
Following the remark we provided figure 1 presenting the schematic overview of NRG1 chromosome position and composition of exons or isoforms
- In section “Detection of NRG1 fusions”, there is a recent method detecting fusion genes using whole-exome sequencing data in cancer patients (PMID: 35251131), would this algorithm help for the detection of NRG1 fusions? Please comment on this point.
Following the remark we are acquainted with the methodological aspects described in the suggested manuscript. Deng et al. proposed the Fuseq-WES pipeline to detect fusion genes from whole-exome sequencing data. In the filtering step for fusion candidate selection, the algorithm focuses only on protein-coding genes. This method discards the intronic sequences and seems not to be sensitive enough to detect the NRG1 fusions since, as we mentioned in the manuscript, “the NRG1 gene consists of large, non-coding fragments that might carry the possible breaking points.”
- All figures in this submission are blurred and the text is difficult to read, the authors should provide higher-resolution images. Ideally, they should use vector graphics, which will maintain their quality and readability across different viewing scales and formats.
Following the remark we uploaded the separate high-resolution figures according to the journal's guidelines.
Minor editing of English language required
Following the remark we provided careful language editing in the final version.
Reviewer 3 Report
Comments and Suggestions for Authors
1. How much Non-small cell lung cancer (NSCLC) is prevalent in European countries.
2. What is the epigenetics role of NRG1 in the pathogenesis of small cell lung cancer (NSCLC)?
3. Does NRG1 have the potential to be used as a biomarker for other cancers?
4. What is the rate of prevalence among different populations having NRG1 alterations?
5. Explain the mechanism and the role of NRG1in NSCLC with figure.
Author Response
Reviewer 3
We are very grateful for the Reviewer’s constructive comments, which have greatly helped us restructure our manuscript. We hope the Reviewer will find our revised manuscript more precise and useful for the readership of Cancers
- How much Non-small cell lung cancer (NSCLC) is prevalent in European countries.
Following the remark in the final version, we provided the precise statement regarding the NSCLC prevalence in Europe:
“In 2020 lung cancer accounted for 11.9% of all new cancer diagnoses in Europe constituting about 480,000 people”
- What is the epigenetics role of NRG1 in the pathogenesis of small cell lung cancer (NSCLC)?
Following the remark we provided a statement regarding the epigenetic role of NRG1 in the pathogenesis of NSCLC:
“Moreover, there are some premises that epigenetic changes may also dysregulate the NRG1 expression leading to its involvement in cancer development and progression”
- Does NRG1 have the potential to be used as a biomarker for other cancers?
Yes, the NRG1 may be a potential biomarker in other cancers, Following the remark we clarified this statement in the final version of the manuscript:
“Thus, the NRG1 fusions may be considered as biomarkers in various cancer types. “
- What is the rate of prevalence among different populations having NRG1 alterations?
Following the remark we implemented in the main text the information about the prevalence of NRG1 fusions in other cancers:
“The prevalence of NRG1 rearrangements in other types of cancers is similar to NSCLC and amounts to 0,5% in cholangiocarcinoma, pancreatic carcinoma and renal cell carcinoma; 0,4% in ovarian cancer; 0,2% in breast cancer and sarcoma”
- Explain the mechanism and the role of NRG1in NSCLC with the figure.
Following the remark we structured the figure caption in a more precise way explaining the mechanism and role of NEG1 in NSCLC:
“In figure 2 we presented the scheme of dimeric ErbB downstream signaling pathways regulated by the NRG1 protein. The NRG1 protein through different biological cascades affects the ErbB dimers for protein synthesis, cell survival, cell apoptosis, control of cell cycle and metabolism, as well as cell migration, invasion or differentiation.”